# Ancient DNA reveals a family ossuary and long-distance migration on the Pacific coast before the Inca Empire

Jacob L. Bongers [1,2,3,14] ✉, Jordan A. Dalton [4,14] ✉, Erik J. Marsh [5], Juliana Gómez Mejía [6], Joshua R. Robinson [7], Jo Osborn [8], Emily B. P. Milton [9], Alexis Rodriguez Yabar[10], Irving Aragonéz Sarmiento[11], Noemi Oncebay Pizarro[11], Kalina Kassadjikova[12] & Lars Fehren-Schmitz [12,13] ✉

This paper tracks long-distance migration on the Pacific coast that began no later than the thirteenth century AD. Genome-wide data for 21 sampled individuals from the lower and middle Chincha Valley of southern Peru show shared ancestry with groups 700 km to the north. A large-scale polity known as the Chincha Kingdom controlled the Chincha Valley from the thirteenth century until the fifteenth century, when it fell to the Inca Empire. The earliest migrants have unadmixed ancestry, whereas in subsequent generations, intermarriage resulted in admixtures from neighboring coastal areas. Relatives buried together in a family ossuary practiced consanguineous endogamy. We build a generation-scale Bayesian model informed by an aDNA-based family tree and individual calibration curves for estimated proportions of marine diet, addressing long-standing difficulties with temporal precision on the Pacific coast due to the marine reservoir effect and uncertainty inherent in estimating marine consumption based on $\delta^{15}$N. These data demonstrate population continuity from the thirteenth to fifteenth centuries, coinciding with persistent traditions of cranial modification and post-mortem red pigment application. We reveal close-knit and far-reaching coastal interaction networks that shaped the sociopolitical landscape encountered by Inca emissaries before they integrated these communities into their empire.

The biological dimensions of cultural transformations are a central topic in anthropology, raising crucial questions about how the development of ancient societies relates to genetic diversity, kinship practices, and interaction at local and broader scales[1–3]. From trade to conquest, interregional interactions often involve population movements and marriage alliances, which foster connectivity among distant groups[4]. Migration is a fundamental part of human behavior and can serve as both a driver for and a consequence of major shifts in

[1]Discipline of Archaeology, School of Humanities, Faculty of Arts and Social Sciences, University of Sydney, Sydney, NSW, Australia. [2]The Vere Gordon Childe Centre, Faculty of Arts and Social Sciences, The University of Sydney, Sydney, NSW, Australia. [3]The Australian Museum Research Institute, Australian Museum, Sydney, NSW, Australia. [4]Department of Anthropology, SUNY Oswego, Oswego, NY, USA. [5]Laboratorio de Paleoecología Humana, Instituto Interdisciplinario de Ciencias Básicas (ICB), Consejo Nacional de Investigaciones Científicas y Técnicas (CONICET), Facultad de Ciencias Exactas y Naturales, Universidad Nacional de Cuyo, Mendoza, Argentina. [6]Department of Anthropology and Sociology, University of Caldas, Manizales, Colombia. [7]Archaeology Program, Boston University, Boston, MA, USA. [8]Department of Anthropology, Texas A&M University, College Station, TX, USA. [9]Department of Anthropology, National Museum of Natural History, Smithsonian Institution, Washington, DC, USA. [10]School of Human Evolution and Social Change, Arizona State University, Tempe, AZ, USA. [11]Independent scholar, Ica, Peru. [12]UCSC Paleogenomics, Department of Anthropology, University of California, Santa Cruz, CA, USA. [13]UCSC Genomics Institute, University of California, Santa Cruz, CA, USA. [14]These authors contributed equally: Jacob L. Bongers, Jordan A. Dalton. ✉e-mail: jacob.bongers@sydney.edu.au; jordan.dalton@oswego.edu; lfehrens@ucsc.edu

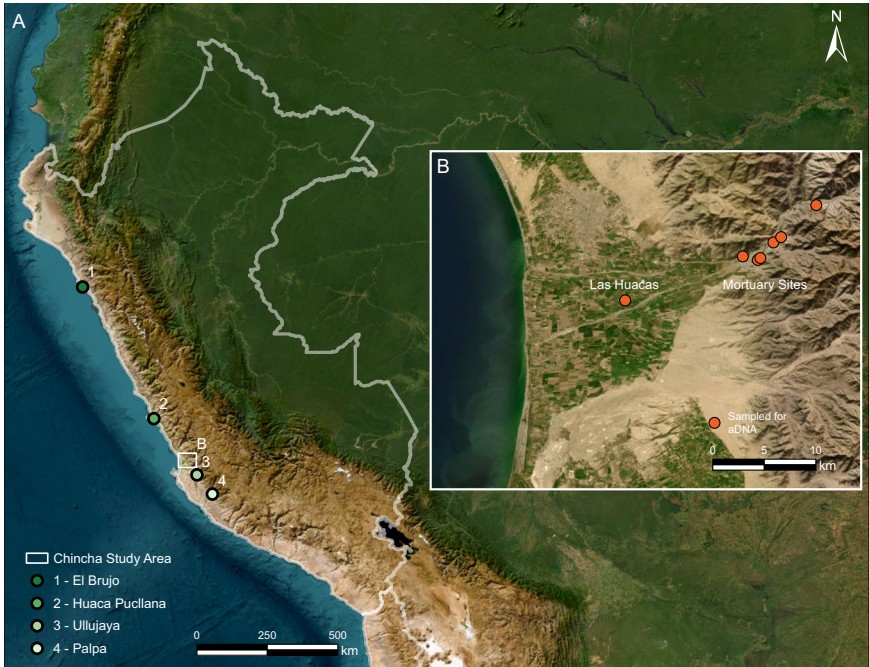

**Fig. 1 | Map of the study area. A** Locations of the Chincha Valley and other Andean sites referenced in this study that yielded ancient DNA data. **B** The archeological sites under investigation for this study. Basemaps for **A**, **B** were obtained from the World Imagery dataset (https://www.arcgis.com/home/item.html?id= 10df2279f9684e4a9f6a7f08febac2a9) and created with ArcGIS Pro v3.6.2. Sources: ESRI, Michael Bauer Research GmbH 2022, Instituto Nacional de Estadística e Informática (INEI), Earthstar Geographics, Vantor.

socioeconomic organization reflected in material culture[5,6]. In the pre-Hispanic Andes (pre-AD 1532), camelid caravans and watercraft played key roles in enduring interaction networks linking kin-based groups and expansionist states alike across ecological zones[7]. This generated a "complex connectivity" reflected in shared technologies, artistic styles, and mortuary traditions[7–9]. Yet important questions remain about how and when interregional interaction occurred *within* ecological zones, and how often the movement of people accompanied the circulation of goods and knowledge.

This study explores these questions on the Andean Pacific coast, a key corridor for movement and trade in South America[10–12] and one of the continent's sustained demographic centers. Here, we combine genome-wide, archeological, and historical data in the Chincha Valley of southern Peru to explore genetic diversity and kinship practices between the thirteenth and sixteenth centuries AD (Fig. 1). This short span was marked by the rise of the local Chincha polity, its integration into the Inca Empire, and population collapse shortly after Spanish invasion in the AD 1530s[13]. Our research shows that both long-distance migration from the north coast to Chincha and genetic exchanges between distinct coastal populations occurred before the Inca conquest. We explore climate, trade, and the expansion of the powerful, northern Chimú polity as potential push–pull factors that might have driven this migration and sustained long-term extra-local contacts. This study also illustrates the central role that biological kinship played in some communities by identifying relatives who practiced consanguineous endogamy and were buried together in a family ossuary.

We assembled genome-wide data from 21 individuals from graves at a large lower valley complex (Las Huacas, $n = 8$) and in six cemeteries distributed across the middle Chincha Valley ($n = 13$). To address the limitations of existing chronologies in coastal Peru[14], we produced a composite Bayesian chronological model that accounts for multiple factors: individual mixed calibration curves based on marine diets, the age of tissue formation, generational gaps based on an ancient DNA (aDNA) family tree, and depositional sequences. This model produced an improved regional marine ΔR and a generational chronology.

This study contributes valuable data to persistent debates in archeology concerning the role of biological kinship in social organization and the relationship between migration, interaction, and cultural diffusion. An important question emerges from our research: how did migration, interregional connectivity, and kinship shape the origins and development of coastal societies and their integration into expansionist states? Local and generational-scale approaches integrating aDNA with multiple lines of independent evidence[3,15] are essential for addressing this question and enhancing understandings of the relationship between human biology and social change within and beyond the Andes.

Sixteenth-century sources describe a powerful coastal polity in the Chincha Valley. This polity's territory encompassed the Chincha and Pisco valleys and possibly parts of the Topará and Cañete valleys to the north[16]. It comprised at least 30,000 male tribute payers[12], indicating a total population possibly exceeding 100,000 people. The Chincha economy followed the model of horizontal economic complementarity, structured around commercial exchange among endogamous groups of producer-specialists[17]. These included at least 10,000 fisherfolk, 12,000 farmers, and 6000 artisans and merchants, each residing in distinct sectors of the Chincha Valley[18]. Merchants acquired silver, gold, emeralds, and other prestige items for exchange with elites across different areas, including the southern Ica Valley[12]. These merchants reportedly traded along the coast and into the highlands, traveling by balsa rafts and llama caravans[12], and likely used balances (scales), which have been recovered throughout the Chincha Valley[19].

The Inca Empire incorporated the Chincha Kingdom in the early AD 1400s[20] forging a rare, negotiated relationship[17]. The current model[17] states that the Chincha voluntarily became a seafaring client to gain privileged access to the prized *Spondylus* trade, which was previously controlled by the Chimú on the north coast. The Chincha clearly held an elevated status in the eyes of the Inca. For example, at the pivotal Cajamarca battle between the Inca and Spanish forces, the Chincha lord sat alongside the Inca Emperor[21].

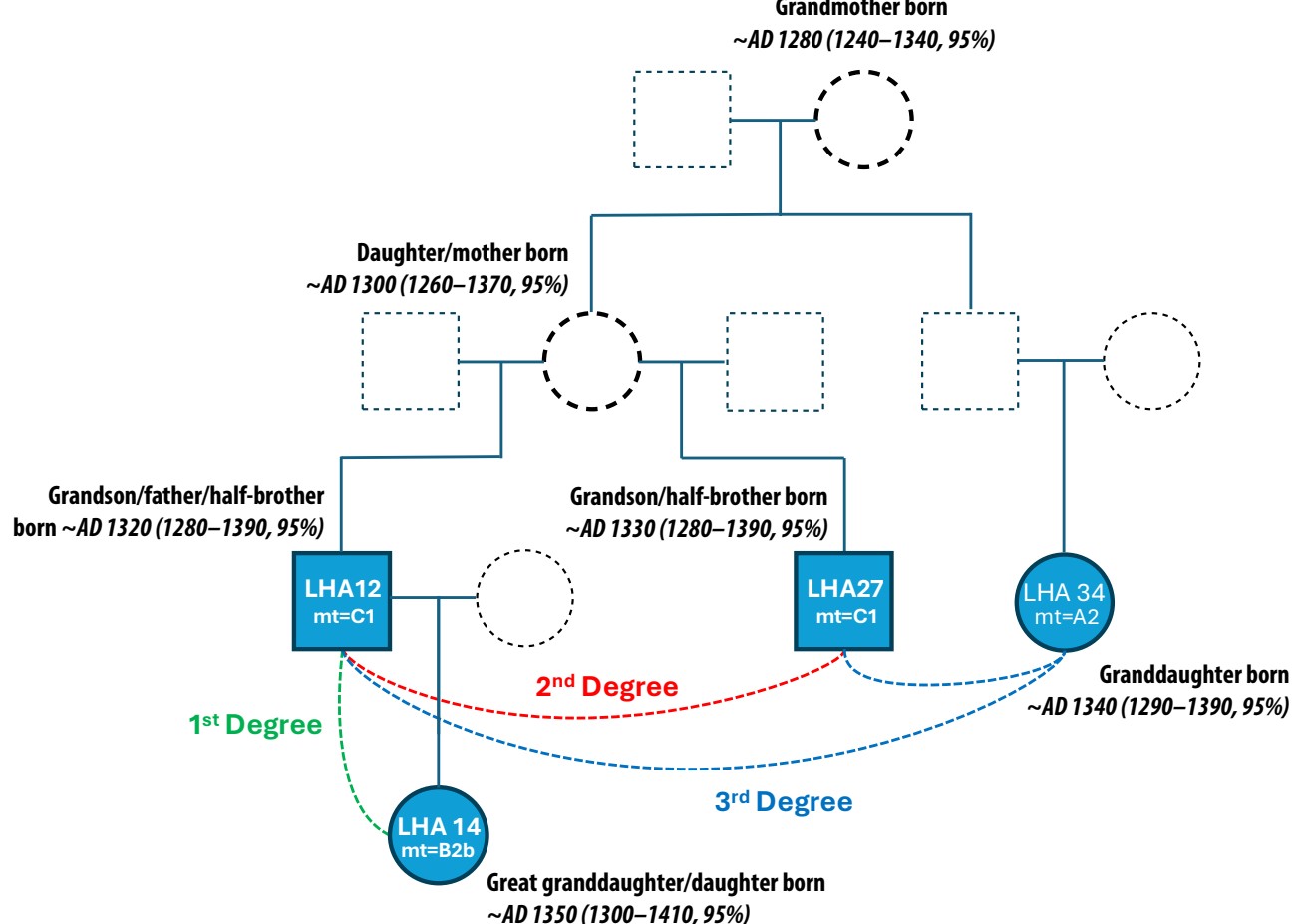

**Fig. 2 | aDNA-based reconstruction of the family tree of four individuals buried in the communal ossuary at Las Huacas.** The dates show the modeled median birth date for each individual (with 95% probability ranges). Lifespans estimates are constrained by long-term global age trends of mothers and fathers. Dates are further adjusted by unique mixed calibration curves that account for each person's marine diet proportions. Dates rounded by 10 years. Circles are females; squares are males. Dotted lines indicate unsampled individuals; two are included in the chronological model (darker dotted lines).

This paper presents genome-wide analyses from Las Huacas and the middle Chincha Valley. The middle Chincha Valley is an example of a *chaupiyunga*, a transitional ecological area situated between the coastal plains and the highland valleys, which is conducive to growing important crops such as coca and maize. Recent archeological research in the middle valley has documented over 500 graves that cluster into 44 mortuary sites[22]. Graves fall into two categories: sub-terranean cists and large, accessible mausolea (*chullpas*), both of which contain multiple individuals[23]. Individuals in cists are in extended positions, whereas *chullpas* feature individuals painted with hematite- and cinnabar-based red pigments, and vertebrae strung on reed sticks ("vertebrae-on-posts") dating to the sixteenth century[22,24,25].

Las Huacas, located in the center of the Chincha Valley's alluvial plain, consists of distinct mound groups separated by modern-day agricultural fields. At 100 hectares, the site was the second largest in the valley. Excavations in a single room (Room A2) of Complex N1 (Supplementary Fig. 1) revealed diverse mortuary features, including subterranean tombs, open-air contexts, individual burials, a large communal ossuary, and extensive secondary burial practices[26,27]. Some of these treatments of the dead, such as the application of red pigment to skeletonized remains and reed-strung vertebrae[28], closely resemble practices documented in the middle valley.

## Results

We successfully extracted DNA for 25 individuals and subsequently enriched for 1,237,207 targeted SNPs across the human genome[29],

using the TWIST Ancient DNA enrichment kit[30,31]. Mitochondrial and X-chromosomal (for individuals with XY karyotypes) contamination rates for most individuals analyzed were below 3%, and the observed damage rates at the read termini were above 3%, indicating authenticity for the obtained genomic data (Supplementary Data 2a). Four individuals (JUC27, JUC34, LHA23, LHA28) exhibited contamination rates above the acceptable threshold and were excluded from further analyses. Combined with six, previously-reported middle valley individuals[15], our final Las Huacas dataset includes the following: 1) six individuals from a large communal ossuary, 2) one primary extended burial, and 3) an unarticulated cranium associated with litter burials[26]. In the middle valley, we have 11 individuals from nine fieldstone *chullpas* and two individuals from separate cists in the middle valley, alongside direct and associated ¹⁴C AMS dates (Supplementary Data 2a). Chromosomal sex estimates indicate that 12 individuals were biological females, and 15 were biological males. The crania of 13 sampled individuals from both study areas were painted with red pigment (Supplementary Dataset 1a), making this one of the first genome-wide studies on individuals who received this treatment.

### Genetic diversity and demography

We determined the biological relatedness between individuals buried in the Chincha Valley sites using the software READv2[32]. In total, we identified one pair of first-degree, three pairs of second-degree, and four pairs of third-degree relatives (Supplementary Data 2b). The main group of relatives were buried together in a family ossuary at Las

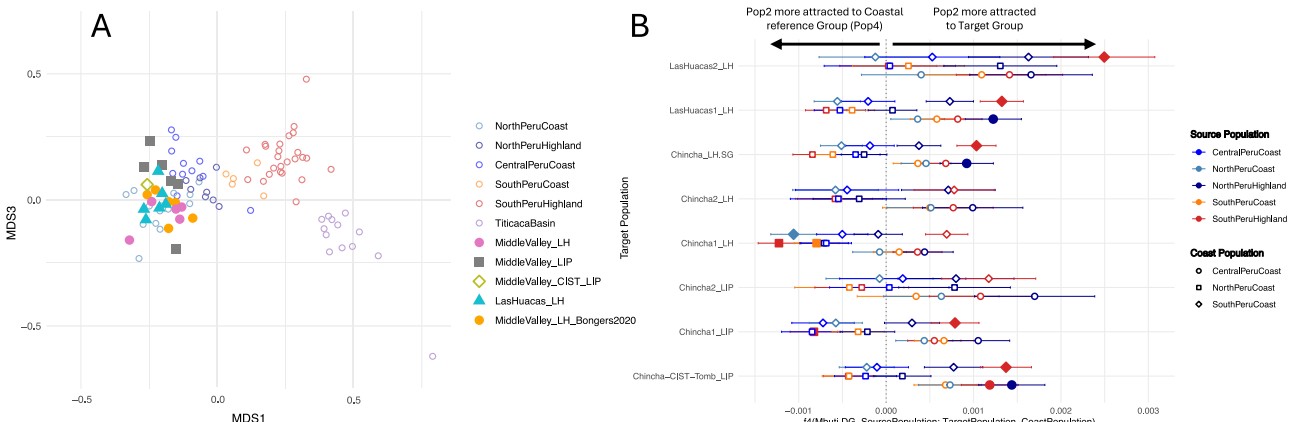

**Fig. 3 | Genetic affinities and allele-sharing patterns of sampled Chincha individuals. A** MDS1×MDS3 plot of an outgroup−f3 distance matrix of the form 1/f3(Ind1, Ind2; Mbuti), where Ind1 or Ind2 is a newly reported or previously published ancient Andean individual. **B** Plot of f4-statistics of the type f4 (Mbuti.DG, SourcePopulation; TargetGroup, PeruvianCoast). Results are presented as F4 estimates ± standard errors. The test evaluates whether Target Group, one of the newly reported Chincha populations (y-axis), forms a clade with any of the three coastal reference groups or whether any of them shares more alleles with other South American source populations. The color of the shapes and error bars indicates the tested source population (Middle Blue = Central Coast, Light Blue = North Coast, Dark Blue = North Highland, Orange = South Coast, Red = South Highland), while the shape indicates which coastal reference was used in the test (Squares = North Coast, Circles = Central Coast, Diamonds = South Coast). Statistically significant test results are shown as filled shapes, while non-significant results are shown as unfilled shapes.

Huacas. By combining the genetic estimates with demographic information such as age and sex, along with their mitochondrial haplotypes, we were able to reconstruct the most likely genealogy for these four individuals (Fig. 2). A mismatch in mitochondrial haplotypes suggests LHA12 was the father of LHA14. The male LHA27 shares a mitochondrial haplotype with LHA12, making him the paternal uncle of LHA14. LHA34 shares a set of grandparents with LHA12 and LHA27. We do not observe any pairs of first to third-degree relatives among middle valley individuals. However, one middle valley individual (JUC30) was a second-degree relative of LHA67, a male buried at Las Huacas in a later disturbed context (Supplementary Data 1a).

The individuals from the middle valley, grouped by chronology, exhibit a degree of genetic diversity, measured in the form of conditional heterozygosity (CH = 0.205 ± 0.001; Supplementary Fig. 2; Supplementary Data 2b), that falls well within the observed average in the Central Andes[33–36]. In the middle valley, CH is slightly higher in later individuals (Supplementary Fig. 2). Conversely, in the lower valley, CH is especially low (CH = 0.201 ± 0.001).

We evaluated Runs of Homozygosity (ROH) profiles for the individuals from the middle valley and those from Las Huacas (reported here and previously published[15]; Supplementary Fig. 3). The latter group exhibits higher proportions of short ROH (4–8 cM) for many of the investigated individuals, suggesting a persistently smaller effective population size for Las Huacas compared to the middle valley individuals[37]. Three individuals from Las Huacas had long 20–300 cM ROH greater than 50 cM, suggesting their lineages had recent close-kin unions (parents being second- or first-degree cousins; Supplementary Fig. 3)[37]. The individuals from the middle valley do not exhibit any long ROH fragments, which is in stark contrast to the observations at Las Huacas and other contemporary sites that indicate a general trend of an increase in closer-kin unions in the Andes starting with the Late Intermediate Period[38].

## Genetic affinities and ancestry

The genetic diversity of pre-Columbian coastal populations in what is today Peru is currently grouped into three populations: 1) the Northern Peruvian Coast, represented by individuals from the Chicama Valley, 520 km north of Lima (AD 200–1200); 2) the Central Peruvian Coast, represented by individuals from Huaca Pucllana in Lima (AD 200–1400); and 3) the South Peruvian Coast, represented by individuals from archeological sites around Ica and Palpa, 340 km south of Lima (AD 200–1300; Fig. 1). Previous studies have found each of the three populations to be genetically continuous during the specific periods for which data were available[15,33], and that this genetic continuity largely persists to this day in those regions[39,40]. All three coastal groups have relatively distinct genetic profiles while sharing a common ancestor after Central Andean coastal and highland populations split around 9000 years ago[33]. The genetic diversity along the Peruvian Pacific coast shows a North–South gradient; its genetic structure is shaped by genetic drift and interactions with non-coastal populations (Supplementary Data 2c, Supplementary Fig. 4).

To explore the genetic relationships of individuals from the Chincha Valley with the aforementioned coastal communities, we calculated Outgroup-F3 statistics of the form f3(Mbuti.DG; AndeanIndividual-X, AndeanIndividual-Y), where "AndeanIndividuals" are previously published ancient genomes from the Central Andes[15,33–36], and performed MDS analyses on the pairwise matrix of the inverted f3 estimates (1/f3). Individuals with less than 20,000 SNPs and first- and second-degree relatives were excluded. The MDS plot (Fig. 3A) replicates the north-to-south gradient in the Central Andes. Surprisingly, both published[15] and this paper's Chincha Valley data[15] cluster with individuals from the North and Central Coast in the MDS plot and not with those from the Central and the South Coast, as might be expected based on the location of the burials. Furthermore, while most of the middle valley individuals buried in *chullpas* appear to drift more towards individuals from the Central Coast (except JUC12), the earlier middle valley individuals buried in cist tombs, as well as most of the later individuals from the middle valley and Las Huacas, are more similar to North Coast individuals.

We used qpWave (v1.200) from ADMIXTOOLS[41] to determine if individuals from Las Huacas or the middle valley form a genetically homogeneous group relative to a set of outgroups. The resulting matrix of pairwise qpWave tests shows that, except for a few outlier pairs, the individuals from each chronological group (LIP-Cist, LIP-Chullpa, LH) and geographic location (middle valley, Las Huacas) form genetically homogeneous groups. Subsequently, we grouped individuals according to the results to strengthen statistical power (Supplementary Data 2a, d).

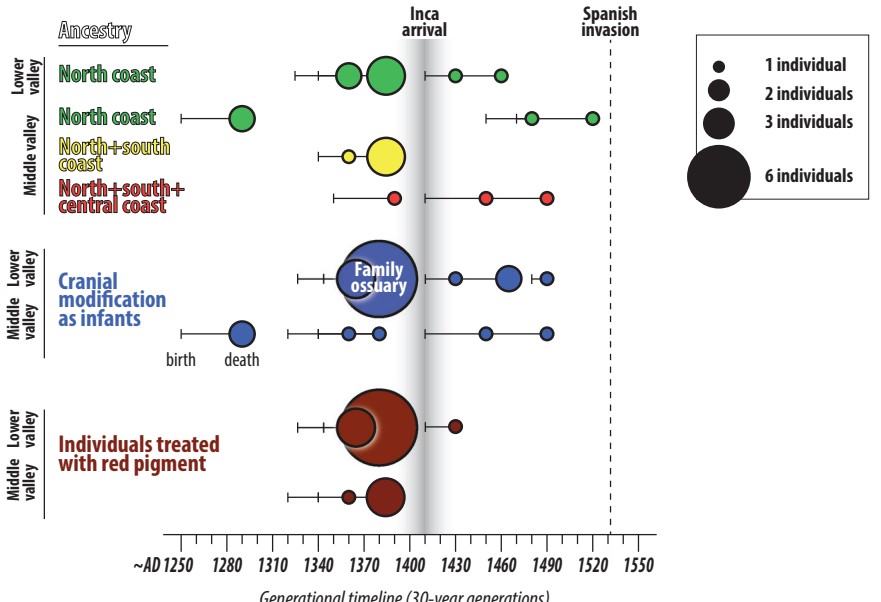

**Fig. 4 | Infographic showing the lifespans and ancestries of Chincha Valley inhabitants.** The data are also grouped according to two practices, cranial modification and the application of red pigment to crania, separated by valley area (lower and middle). Lines and colored circles indicate the median birth and death date, respectively; probability distributions are not shown for simplicity's sake. The distributions are non-Gaussian; the average length of the 95% uncertainty range is 132 years, comparable to an error of ±33 years. Some individuals appear in multiple plots; for example, all individuals from the family ossuary with aDNA data have north coast ancestry. All dated individuals from the family ossuary have both cranial modifications and red pigment.

To explore whether any of Chincha groups are cladal with any of the three Peruvian coastal populations, we calculated F4-statistics (Mbuti.DG, AndeanRegion; ChinchaPopulation, PeruvianCoast), where "PeruvianCoast" refers to either the ancient North, Central, or South Coast reference populations, and "AndeanRegion" represents the different regional genomic ancestries defined by Nakatsuka et al.[33]. The earliest individuals from the middle valley (Peru_Chincha-CIST-Tomb_LIP), the middle valley individuals reported in ref. [15] and the individuals buried in the Las Huacas ossuary (Peru_LasHuacas1_LH) are only cladal with the Peruvian North Coast Group (Fig. [3]B, Supplementary Data 2e). Using qpWave with the same outgroups and model parameters reported by Bongers et al.[15], all three groups can all be modeled as consistent (1-wave, $p > 0.05$ for rank 0) with North Coast Ancestry (Supplementary Fig. 5, Supplementary Data 2e) with 1-wave of ancestry when compared to each other ($p = 0.25–0.35$).

For this paper's two middle valley groups (Peru_Chincha1_LIP, Peru_Chincha1_LH), non-significant test results are only observed in the f4-tests when paired with the Central Coast group. Meanwhile, Peru_Chincha1_LIP shares more alleles with individuals from the Southern Highlands than with those from the North Coast, but significantly fewer than those from the South Coast. Similarly, Peru_Chincha1_LH shares more alleles with southern populations than the North Coast group but also shares more alleles with the North Coast than the South Coast population (Fig. [3]b, Supplementary Data 2e). Nearly all models testing whether the middle valley population (Chincha1_LIP) could be explained by 1-wave of ancestry from either the Central Coast reference population or any of the Chincha groups returned p-values below the significance value ($p = 0.05$; Supplementary Data 2f). This indicates that none of the models provided a good fit, except for LasHuacas2_LH. Using qpADM[41] to test two-way admixture models employing a rotating model approach[42], we find that, out of 55 tested models, the only one supported considers Chincha1_LIP to be admixed between 80% North Coast ancestry and 20% South Coast ancestry (±9%, $p = 0.186$; Supplementary Data 2f).

This paper's later middle valley group (Chincha1_LH) cannot be modeled as consistent with 1-wave of ancestry when paired either with any of the coastal reference groups or the earlier middle valley group (Chincha1_LIP; Supplementary Data 2f). Using the previously described qpADM approach, this time including Chincha1_LIP, we find only one supported model out of 66, which assumes Chincha1_LH to be admixed between 61% Chincha1_LIP ancestry and 39% Central Coast associated ancestry, although with a very high standard error (±24%, $p = 0.959$; Supplementary Data 2f).

We observe no significant statistics for the F4-tests for Peru_Chincha2_LIP and Peru_Chincha2_LH when compared to any of the coastal populations (Supplementary Data 2e). Additionally, 1-wave qpWave models are supported ($p = 0.06–0.61$) for both groups when paired with either the North Coast or the Central Coast population. However, these results are potentially biased due to relatively low genomic coverage, which results in a lack of statistical power.

The later period individual from Las Huacas (LHA67, Peru_LasHuacas2_LH), who is a second-degree relative to one of the middle valley individuals, appears cladal with both the North- and Central-Coast populations, but shares fewer alleles with individuals from the Southern Highlands than individuals from the southern coast (Supplementary Data 2e). Modeling for Chincha2_LH is consistent with Central Coast ancestry in qpWave ($p = 0.26$; Supplementary Fig. 5, Supplementary Data 2e).

## Radiocarbon dating

**Local ΔR and improved precision.** We estimate that the marine reservoir effect (ΔR) for this time and place was $-314 \pm 52$ years. This is a posterior Bayesian estimate based on all model constraints, as opposed to the conventional approach of using modern shell data. The ΔR is based on a unique mixed calibration curve for each individual, which is proportional to their marine diet contributions. The revised ΔR makes it possible to recalibrate other dates from the region with greater precision. This paper's precise results depend on the constraints built into the family tree (Fig. [2]) as well as a detailed consideration of the depositional sequence. Without a Bayesian model that takes these factors into account, the influence of marine diet

would make individual date calibrations too imprecise to address generational-scale questions.

**Migration was pre-Inca**. Most individuals with north coast ancestry lived well before the Inca Empire was in contact with the valley (Fig. 4). Hence, we can refute suggestions that Inca policies were responsible for initiating these migration patterns. The Inca may have reinforced or formalized preexisting family ties to the north coast, especially since none of our sampled individuals shows any genetic connection to the highlands. Moreover, it is possible that north coast migrants arrived in a relatively unpopulated valley, as there is a paucity of sites and dates from AD 200–1200[43].

In the middle valley, there is a sequence of increasing genetic diversity over time (Fig. 4). Two of the earliest individuals (JUC35 and JUC73) from cist graves have modeled median death dates of -AD 1290. JUC35 has unadmixed north coast ancestry, and JUC73 likely represents the same individual as JUC61, who also has north coast ancestry (Supplemental Information). Next are four individuals with admixed ancestries from the north and south coasts (Peru_Chincha1_LIP), whose median death dates are -AD 1360–1390. Three individuals have admixed ancestries that include the north, central, and southern coast (Peru_Chincha1_LH), with median death dates of -AD 1390–1490. Probability distributions overlap, but the tendency suggests more admixture over time. All individuals have some North Coast ancestry, perhaps via other migrants in the Chincha Valley. Intra-valley connections are suggested by a second-degree relationship between two people from the lower (LHA67) and middle valley (JUC30), who both died in the last generation before the Spanish invasion (Fig. 4).

**Chronology of burial landscapes in the Chincha Valley**. We grouped the deaths in the middle Chincha Valley as a uniform phase, which suggests that burials began -AD 1260 (1180–1320, 95%). Around the same time, the occupation of Complex N1 at Las Huacas began -AD 1260 (1220–1280, 95%), with modeled dates based on the stratigraphic sequence of the Main Room (A2). Conventional chronological treatments have slotted these contexts into temporal boxes such as the Late Intermediate Period or the Late Horizon. Our bottom-up chronology uncovers the weaknesses in these traditional block schemes, in an effort to move toward generational-scale chronologies that can address lived human histories.

**Chronology of the Las Huacas family ossuary**. Our results suggest that at least two generations of a family group were deposited in a shared ossuary (Feature 17) at Las Huacas. The oldest sampled burial in the ossuary was LHA12, a man who lived from -AD 1320 (1280–1390, 95%) to -AD 1360 (1310–1420, 95%), estimated from the Bayesian model. Overall, our model estimates that individuals buried in this ossuary died between -AD 1330 (1300–1370, 95%, First) and -AD 1440 (1400–1490, 95%, Last). The family tree includes the grandparents of three individuals (Fig. 2). The unsampled grandparents may have also lived at Las Huacas or played a role in the family's initial migration; the model suggests the grandmother was born -AD 1280 (1240–1380, 95%), around the same time as the earliest burials in the middle Chincha Valley. The chronological model is strongly anchored by the generational gaps between the four family members that can be placed on the family tree (Fig. 2).

## Discussion

While it remains possible that the Inca resettled some people from the north coast to Chincha during the fifteenth century[15], our data show that the initial arrival of northerners clearly predates Inca conquest. The middle valley cist graves indicate that migrants from the north had arrived in Chincha by at least the thirteenth century. The long-distance migration of northerners to Chincha is strongly supported by previous historical, archeological, and genetic data. Pedro Cieza de León, a

sixteenth-century chronicler, describes the earliest Chincha people as a group led by a valiant captain who came from afar to conquer the valley[44]. Central- and north-coast-style ceramics and textiles have been identified throughout Chincha[45,46], including in middle valley tombs previously sampled for genome-wide analyses that identified north coast ancestry[15].

Push–pull factors may explain this migration. Climate hazards on the north coast, such as the El Niño-Southern Oscillation (ENSO) phenomenon, could have pushed residents to migrate. An expanding, powerful north coast polity such as the Chimú, with their strong sea-faring tradition and management of over one thousand kilometres of Pacific coastland, could have brought colonists to Chincha. People may have also been seeking to escape Chimú expansion. Alternatively, north coast traders could have migrated to Chincha to secure access to local sources of seabird guano[47] and copper as well as the cinnabar mines of Huancavelica[48].

Genetic admixtures from the north, central, and south coasts identified in the middle valley are consistent with this scenario: northerners migrated to Chincha and then intermarried with distinct communities. This is significant because it suggests that colonial-era marriage alliances, discussed below, built upon coastal connections established prior to the Inca conquest. Shortly after the Spanish conquest, marriage alliances formed between families of similar social standing on the central and northern coasts[10]. For example, women from fishing settlements in the Pachacamac Valley often married men from neighboring coastal towns and more distant places such as Santiago de Cao on the north coast[49]. This helps explain elites with Mochica and Quingnam surnames in the Lima region and of Quechua speakers from the central coast in parts of northern Peru[10]. Notably, highland families were excluded from these coastal marriage networks[50], a pattern also documented during the Colonial Period on the Chilean coast[51].

Other social processes may help explain these genetic admixtures. Chincha's interactions with the Pisco Valley and Ica Valley elites[12,13] may account for gene flow from the south coast. Colonial sources document both inter- and intra-valley marriages among south coast elites[52]. The powerful oracle of *Chinchaycamac*, considered a "child" of the Pachacamac oracle from the central coast[11], was established in Chincha at this time. As a religious center that received offerings and likely pilgrims from afar, *Chinchaycamac* may have also facilitated genetic exchange.

All sampled individuals, however, have some degree of north coast ancestry, demonstrating population continuity from the thirteenth to the fifteenth centuries. This continuity coincides with shared mortuary practices and cranial modification between Las Huacas and the middle valley (Fig. 4). Postmortem use of red pigment on crania[25] and tabular cranial modifications, produced by boards and bindings applied in infancy[53], are documented on the north coast[54,55] and among sampled individuals in the Chincha Valley who have median birth dates between the thirteenth and fifteenth centuries (Fig. 4; Supplementary Data 1a). In the middle valley, these practices appear on unadmixed and admixed individuals that were placed in both cists and *chullpas*. At Las Huacas, they are documented across burial contexts, including an ossuary where related individuals were buried together, providing strong evidence of a shared group identity.

Genetic evidence from Las Huacas offers a fine-grained perspective on local social organization, suggesting that biological kinship played a central role. The remains of a family group were buried together in the Feature 17 ossuary. The lower genomic diversity, increased rate of short and long ROH, and the observed pairs of relatives suggest that this group drew their members from a reduced mating pool for several generations, consistent with practices of consanguineous endogamy. This contrasts with the individuals buried in middle valley sites who recruited their mating partners from a larger, more diverse pool. The Las Huacas data align with broader shifts

towards kin-based mortuary practices[56] and an increased rate of close-kin unions from the twelfth to sixteenth centuries[38]. Other aDNA studies of late pre-Hispanic human remains in the Andes also document the interment of closely related individuals within the same mortuary structures[57,58].

The close familial relationships identified at Las Huacas may indicate the presence of a corporate entity, such as an *ayllu* or *parcialidad*. These are descent-based groups, often subordinated to one or more lords and organized around principles of reciprocity, redistribution, and real and fictive kinship[56,59]. Within these arrangements, endogamy may have served as a strategy for retaining control over resources within the group[59]. These results provide support for Andean models of socioeconomic organization that are structured around endogamous groups of producer-specialists, a pattern that appears applicable to some segments of the Chincha population as well as other coastal societies[12,18]. While the paleogenetic results support the importance of familial relationships within the Chincha Kingdom, understanding how the familial unit operated in social and political life will require further research. Importantly, the differences in mating patterns across individuals included in this study may reflect distinct marriage strategies between lower and middle valley inhabitants, but additional samples are needed to evaluate this claim.

## Methods

### Ethics statement

This research was carried out in collaboration with descendant communities and governing agencies in Peru. Fieldwork, exportation of samples, and laboratory analyses were conducted under permits issued by the Peruvian Ministry of Culture. For the middle valley, permits were granted in 2013 (206-2013-DGPC-VMPCIC/MC), 2015(218-2015-DGPA-VMPCIC/MC), 2016 (107-2016-VMPCIC-MC), 2017 (145-2017-DGPA-VMPCIC/MC), and 2018 (148-2018-DGPA-VMPCIC/MC). For Las Huacas, permits were granted in 2017 (001379-2017/DGPA/VMPCIC/MC) and 2019 (035-2019-VMPCIC-MC, 101-2019-VMPCIC-MC). This project emerged from long-term, collaborative research programs (2012–current) involving archeological fieldwork among archeologists and university students from Peru and the United States, as well as community members from the Chincha Valley. This study was fully authorized by the Peruvian Ministry of Culture. We complied with all legal and ethical norms for the study of aDNA and will continue to work with local leaders and museums to share our research findings with communities and incorporate their questions into further research projects[60,61].

### Sample processing

All samples were processed at the Keck Carbon Cycle AMS facility, Earth System Science Department, University of California, Irvine. For organic samples, pretreatment was acid-base-acid (1N HCl and 1N NaOH, 75 °C) prior to combustion. Bone samples were decalcified in 1N HCl, gelatinized at 60 °C and pH 2, and ultrafiltered to select a high molecular weight fraction (>30kDa). $\delta^{13}C$ and $\delta^{15}N$ values were measured to a precision of <0.1‰ and <0.2‰, respectively, on aliquots of ultrafiltered collagen, using a Fisons NA1500NC elemental analyzer/Finnigan Delta Plus isotope ratio mass spectrometer.

### Diet estimates

$\delta^{13}C$ and $\delta^{15}N$ of human dentine and bone collagen samples were used in Bayesian mixing models through the MixSIAR package[62] in R Studio 4.2.3 in order to estimate the proportions of potential dietary source contributions—$C_3$ plants, $C_4$ plants, terrestrial fauna, and marine fauna—to each individual consumer (see Supplementary Information and Supplementary Data 3). To allow the model to estimate the variance-covariance matrix associated with the traces for each individual source, we supply 'raw' as opposed to 'summary' (means and standard deviations) source data. This is a more ecologically realistic scenario, as the 'summary' format must assume independence of all tracers. MixSIAR uses a Markov Chain Monte Carlo (MCMC) model-fitting algorithm. We considered chains converged when indicated by a Gelman-Rubin diagnostic value of <1.05 for all variables.

### Isotopic mixing models

Source categories for mixing models were constructed by compiling published data from the South American Archeological Isotopic Database (SAAID)[63] and combining with new maize data from Chincha[47], marine fauna data in the form of guano birds from Jahuay[47], and plant data (both $C_3$ and $C_4$) from Las Huacas[64]. SAAID data were first screened to include only samples that may be considered temporally and geographically consistent with human data. Specifically, only samples that reported both $\delta^{13}C$ and $\delta^{15}N$ values from Peru chronologically attributed to the Late Intermediate Period, the Late Intermediate Period/Inka Period, and Late Horizon are included. From these, faunal remains, representing camelids, cuy, deer, and marine fauna[65–71], were further restricted to bone collagen samples (to control for dietary offsets) and screened for appropriate quality control metrics (atomic C:N ratios between 3.1 and 3.5, %C > 20%, and %N > 10%)[72]. Samples that did not report quality control metrics were excluded. Taxa unlikely to be dietary, such as canids or unidentified birds, were also removed.

The $C_3$ plant source category includes the geographically and temporally screened samples identified using the SAAID[66,73] as well as nine samples from Las Huacas[27]. The $C_4$ plant category consists exclusively of maize cob samples from the middle Chincha Valley and Las Huacas[64]. While three maize samples in the SAAID pass our screening requirements, they are identified as seeds, and we exclude them to avoid issues related to differential uptake across plant parts[74]. Terrestrial fauna includes all SAAID samples that pass screening checks. Marine fauna consists of 11 guano birds from the site of Jahuay[47] and 1 sea lion sample from SAAID that pass the screening requirements. While these categories may aggregate particular plants or animals that were more or less important to human diets and prevent the determination of an estimated contribution for a particular plant or animal source, it minimizes the potential for underdetermined models whereby all (or nearly all) sources return an equal estimate of dietary contribution. Following other studies that have constructed human diet mixing models in South America[75], we use the following trophic enrichment discrimination values:

$\Delta\delta^{13}C_{human-terrestrial}$: 1.0 ± 0.0‰
$\Delta\delta^{13}C_{human-marine}$: 1.0 ± 0.0‰
$\Delta\delta^{13}C_{human-C4}$: 4.8 ± 0.5‰
$\Delta\delta^{13}C_{human-C3}$: 4.8 ± 0.5‰
$\Delta\delta^{15}N_{human-terrestrial}$: 3.0 ± 0.0‰
$\Delta\delta^{15}N_{human-marine}$: 5.5 ± 0.5‰
$\Delta\delta^{15}N_{human-C4}$: 3.0 ± 0.0‰
$\Delta\delta^{15}N_{human-C3}$: 3.0 ± 0.0‰

As there are no straightforward reasons to weigh the importance of these dietary source categories differentially, we apply an uninformative/generalist prior to the model. This means the model is largely driven by the distribution and variance in source $\delta^{13}C$ and $\delta^{15}N$. To model an ecologically realistic scenario, we apply process errors that account for individual specialization within a group/population and sampling error. Effectively, this allows consumers within the model to operate somewhere between perfect specialists and perfect integrators (as opposed to being one or the other) and creates a better fit between narrow consumer data and wide source data[62]. Dietary mixing models are run in the MixSIAR R package[62] utilizing the package's common functions without any customized script. We include the running of the MixSIAR package within our customized Mix-Cal-Lot script[76] (described below) that allows for the output of MixSIAR to be translated for use with OxCal.

For 10 individuals from Las Huacas, there are independent dietary isotopes from vomer samples, which will be presented in a future publication. For the purposes of this paper, we compared dietary estimates taken from other dated tissues, mostly teeth (Supplementary Data 1). Dietary estimates were very similar in almost all cases. We also compared calibrated radiocarbon dates using both estimates of marine diet and the results were essentially identical. The minor differences between samples from the same individual could be useful in building life histories for future publications.

## Radiocarbon dates, calibration, and modeling

This paper models radiocarbon dates from 43 carbon samples that were exported with the following permits for the middle valley (206-2013-DGPC-VMPCIC/MC, 218-2015-DGPA-VMPCIC/MC, 107-2016-VMPCIC-MC, 145-2017-DGPA-VMPCIC/MC, 148-2018-DGPA-VMPCIC/MC) and Las Huacas (001379-2017/DGPA/VMPCIC/MC, 035-2019-VMPCIC-MC, 101-2019-VMPCIC-MC. Radiocarbon dates were processed at UCIAMS and NOSAMS. UCIAMS samples on human tissues met quality control standards for well-preserved collagen, with atomic C:N ratios of 3.1–3.4[72].

Terrestrial samples were calibrated with SHCal20[77], since this region is probably outside of the area influenced by Northern Hemisphere air mixtures[78]. Marine samples were calibrated with Marine20[79]. For samples on human tissue, we mixed the terrestrial and marine calibration curves. A unique curve mixture was built for each individual, in the same proportions as their marine protein intake. We did not use a marine ΔR based on modern shells but instead let the ΔR float freely in the Bayesian model.

Based on the sampled human bone tissues, which included tissues that formed at different points in an individual's life, and osteological age-at-death estimations, we estimated birth and death dates for each individual. For the family members found at Las Huacas, we built in expected generational gaps, modeled here as the average age at a child's birth for mothers (24 ± 6) and fathers (28 ± 7). We also incorporated a detailed reconsideration of the depositional sequence at Las Huacas.

All calibrations and temporal relationships were combined into a single model run in OxCal 4.4[80]. Results are rounded by 10 years, and italics denote results from Bayesian models. Since calibrated dates are non-normal distributions, we report medians (-) and 95% probability ranges.

## Building mixed marine–terrestrial calibration curves with non-normal probability curves

It is standard practice to use marine diet estimates to adjust radiocarbon dates, since marine protein affects the carbon content of the consumer's tissues[81]. The extent of this impact is usually based on a rough estimate of marine protein using linear mixing models, two end-members, and an error term of ±10%, since "very roughly, a change of 1‰ corresponds to about 10 percent change in marine intake"[82]. This is a frequently applied rule of thumb[15,83–85]. However, with the widespread use of Bayesian mixing models, marine dietary estimates can be much more nuanced and are certainly not linear or normally distributed. Using means and standard deviations[86,87] assumes normally distributed data, which is rarely the case. This can have an outsized impact on some radiocarbon dates, as the marine dietary estimate can shift a calibration by centuries.

To address this, we developed a new method to extract the probability density function for marine diet percentage from MixSIAR outputs, a script we call Mix-Cal-Lot[76]. This output is a more complete description of the amount and likelihood of marine diet contributions. The script works with MixSIAR's outputs (updated in v3.1.13), which are checked for normality with a Kolmogorov-Smirnov test. As expected, all 30 individuals in this paper have non-normal marine diet estimates

($p < 0.05$), meaning we should not attempt to describe these distributions with means and standard deviations. The Mix-Cal-Lot script provides two key operations that allow for MixSIAR dietary mixing model estimates to be translated into a format that can become useful for input to OxCal. First, Mix-Cal-Lot extracts and saves the full and original JAGS file (the Bayesian statistical model's 'blueprint') produced during the running of MixSIAR. Standard outputs of MixSIAR do not include the JAGS file, as most users are simply interested in the final dietary estimates and visualizations. Second, since the diet estimates are non-normal probability curves (as shown by the Kolmogorov-Smirnov tests), Mix-Cal-Lot uses the JAGS file to access each individual's MCMC chains to extract values from under the posterior density function of the estimated contribution of marine resources to diet for creating a histogram with 100 evenly sized bins.

While more bins would create a finer-grained histogram, this has a negligible impact. Using these values, we reproduced the same histogram in OxCal 4.4[80] as a probability array with 100 values. In the example below for individual JUC73, the set of 100 values is in bold and underlined, starting with 140 and ending with 1. A unique calibration curve is built for each individual, mixing the terrestrial curve SHCal20[77], appropriate for this part of South America[78], and the marine curve Marine20[79], modified by the local ΔR. The relative contribution of the marine curve is the same as the marine diet contribution described as a histogram. The Mix-Cal-Lot script automates this process, extracting marine histograms for each individual and building the OxCal code. This example shows how we calibrated the date for individual JUC73:

```
Curve("SHCal20","SHCal20.14c");
Curve("Marine20","Marine20.14c");
Delta_R("LocalMarine",U(0,100));
Mix_Curves("Mix for JUC73", "SHCal20", "LocalMarine",
P(−1,101,[0,140,208,211,188,177,148,166,131,125,121,110,98,112,88,
76,69,75,72,57,70,55,49,45,40,30,31,28,34,27,21,21,19,17,17,11,6,12,
13,7,6,8,7,4,8,2,4,3,0,1,4,5,0,2,2,2,2,0,2,1,0,0,1,0,1,1,1,0,3,0,0,0,0,
0,0,0,0,1,0,0,0,1,0,0,0,1,0,1,0,0,0,0,0,0,0,0,0,0,0,0,1,0]));
R_Date("UCIAMS-270835, JUC73, Sector B, Tomb U1",895,20);
```

We ran a sensitivity analysis to assess the impact of different mixing models on the calibrated dates, based on two dietary models with slightly different source data. The two models resulted in lower and higher marine diet estimates. We calibrated birth dates for all 30 individuals with both models and compared the results. The median dates were very similar: using the model with higher marine estimates, medians were 10–30 years younger. The 95% probability curves are shifted a few decades toward younger ages, and a few have long tails that extend into the Colonial Period. Despite these differences, the two diet models produce highly similar calibrated dates. Given the other uncertainties built into each step of this process, it seems unproductive to further consider multiple dietary models. This paper is based on the dietary source data that produce lower marine estimates.

## Marine ΔR for central Peru: constraining dates based on stratigraphic relationships

Consuming marine protein can significantly affect radiocarbon dates, but the influence depends on the proportion of marine carbon in the diet. The marine reservoir effect (ΔR) can be highly variable over time, especially in areas with strong upwellings such as the coast of Peru.

For this reason, calibrating dates must account for ΔR, which is the difference between local and global surface marine radiocarbon, as estimated by Marine20[79]. Typically, this is done with modern ΔR data from a continually updated database[88]. Most of these data points are based on $^{14}$C ages run on shells with known collection dates. Many of these samples are from museum collections from the nineteenth and

twentieth centuries. Hence, these data are most applicable for these centuries.

For the region around the Chincha Valley, the database returns highly variable ΔR estimates, based on 58 [14]C ages on six shells from four locations (8–14°S): Puerto Salaverry, Salaverry, Callao Bay, and Paracas (Supplementary Fig 7)[89,90]. All were collected AD 1908–1948. When using this database, multiple ages from a single location can be accessed by clicking on the underlined map number in the table (note that only one value per location is used by the website's calculation option). For the 58 ages on six shells from this area, ΔR is highly variable, from −181 to +219 years. All six of these shells are all from the early twentieth century and only lived a few years, but ΔR estimates vary by centuries, even when comparing dates from a single shell (Supplementary Fig 7)[89–91]. The standard approach is to create weighted averages for each location, which are similar: −15 ± 59 (all 58 dates), −32 ± 58 (30 dates on two shells from Paracas), and 14 ± 49 (four dates, one shell from Salaverry). Weighted averages may not be appropriate here, as they tend to mask internal variability.

This approach assumes the data points are independent, but often they are from the same short-lived shell. Samples taken along transects of a shell's growth line should be very similar and in sequence, but this is often not the case. This approach cannot account for outliers. Finally, in this region, it is quite unlikely that the near-modern ΔR was the same in the past. Trial models with modern ΔR estimates did not converge; that is, the modern ΔR significantly disagreed with the model's other, more reliable constraints: stratigraphy and generational intervals. Hence, we decided against using ΔR estimates based on twentieth-century data, though this approach can be effective in other regions.

Instead, we followed Marsh's approach in letting the ΔR float freely as a uniform distribution[92,] based on Jones et al.'s earlier suggestion for calculating local, context-dependent ΔRs[93]. This approach treats ΔRs as posterior results instead of prior assumptions. With a free-floating Bayesian ΔR, the model converges on a ΔR that agrees with the rest of the model's constraints. For example, in two archeological contexts from coastal Peru around AD 500, results were similar: −272 ± 37 years, based on two tombs from Huaca de La Luna, and −270 ± 72 years, based on generation gaps between the Señora de Cao burials[92]. This strongly negative ΔR could be the result of "(1) deepwater upwelling reduction generated by extended El Niño conditions, and/or (2) greater than modern El Niño frequency causing [14]C enrichment of surface water"[91]. Importantly, these estimates are based on human tissue formation, which spans at least a few years and sometimes decades. In contrast, the samples from six modern shells in this region seem to capture annual or even seasonal growth, a narrower temporal scale that shows highly variable carbon reservoirs. This mismatch in the temporal resolution and the dynamic upwelling makes shell data inappropriate for comparing to archeological ages from coastal Peru. This is salient in regions such as the Pacific coast with strong deepwater upwelling that can vary strongly by season and/or with strong El Niño activity.

A free-floating ΔR did not converge in a trial model with only dates from the middle Chincha valley, since these contexts do not have other constraints. In contrast, the stratigraphic sequence at Las Huacas allowed the model to converge on a Bayesian ΔR estimate of −314 ± 52 years. This is based mostly on 1) the expected lapses in the family tree and 2) stratigraphic relationships from Las Huacas. This ΔR is then applied to other dates without other constraints, including dates from the middle Chincha Valley. The model assumes a single ΔR value for all samples; it is insensitive to minor changes over the span of dates in the model. Based on current data, the strong agreement indices suggest there were no major temporal differences in ΔR over this span. Individuals with more marine protein in their diets will have a greater influence on the ΔR.

Coastal Peru now has three Bayesian ΔR estimates with medians of −270 and −272 years around AD 500 and −314 years, based mostly on individuals who lived between AD 1300 and 1500 (data are concentrated in the late AD 1300s). All three ΔRs are much more negative and more precise than the data from six shells collected AD 1908–1948. We recommend that future models use modern shell data with caution, especially in Peru, and instead build floating Bayesian ΔR estimates based on archeological priors.

## Estimating birth and death dates based on tissue formation age and age at death

As radiocarbon ages become more precise, error ranges can be less than an individual's lifetime. Some ages are obtained from late-forming tissue that is continually remodeled, for example, ribs, which reflect a date close to an individual's death. In contrast, radiocarbon ages obtained from teeth reflect tissue formed during childhood and adolescence. We use the standardized schedule of tooth formation to place these radiocarbon ages in each individual's lifespan, following Millard et al.[94], who use AlQahtani et al.'s[95] ages of tooth formation. Lane and Marsh[20] recently applied this same approach to the burials at Machu Picchu. Here, we disregard secondary dentine because it only adds 2–5 years to the formation age and only affects older individuals. There are four ages from non-tooth tissues, from the vomer, sphenoid, and vertebrae. In the case of the vertebrae, the age of tissue formation is unimportant because this individual died around 5–7. For the vomer and sphenoid, tissue formation age is not well defined, so here we use an informal approximation of 15 ± 5 years, since the vomer fuses in early adulthood[96]; this can surely be improved upon.

As an example, we continue with individual JUC73. His second right molar was sampled, which formed at age 11 ± 1, so his birth was 11 ± 1 years before the radiocarbon date, reflected in this OxCal code:

Date("Birth JUC73", R_Date("UCIAMS-270835, JUC73 ",895,20)-N(11,1));

Osteological analysis suggests he died between the ages of 40 and 55, so we can add this uniform probability range to his birth date, reflected in this code:

Date("Death JUC73", Date("=Birth JUC73") + U(40,55));

The posterior probability distributions for individual birth and death dates are shown in Supplementary Fig 8.

## Constraining birth dates based on family relationships

Radiocarbon ages can also be constrained by adding priors based on family relationships and the likely age of parents when children are born, for example, mothers tend to be 24 ± 6 years old when they give birth (to any of their children, not only their first), a normally-distributed range from global historic and modern data[20,92]. Fathers tend to be 28 ± 7 years old, though data are sparser.

These estimates are from a global AD 1948–2015 dataset[97]. In this dataset, the ages are grouped in five-year categories. Trial subsamples from a variety of decades and countries showed very minor variation in the mean and standard deviation, often by around one year. These data can be robustly fitted to very similar Gaussian curves. Data trends from the United Nations Statistics Division[98] show the same pattern with more precise data from the last twenty years. Despite minor increases in recent decades, the long-term global average of 24 ± 6 is quite stable, and differences are negligible when applied to radiocarbon dates, since calibrated estimates are usually rounded by 10 years. This estimate for mothers broadly agrees with Fenner's[99] cross-cultural survey, which suggests an average of 27.3 years for mothers in industrialized nations and 25.6 years among hunter-gatherers. Alternatively, averages can be estimated with genetic mutation rates over the last 250,000 years[100], which suggest similar figures of 23 ± 3 years for mothers and 31 ± 5 for fathers.

In this paper's sample, only four individuals have clear relationships we can confidently place in a family tree. Three had the same

grandparents (LHA12, LHA27, and LHA34); two are half-siblings (LHA12 and LHA27). One of them (LHA12) had a daughter (LHA14) (see Supplementary Data 2b).

Here, we built a model starting with the cousins' grandmother, since using mothers has a better-defined estimate for age at birth. We could do the same with the grandfather, or with estimated lapses between deaths, but both of these approaches are less precise. When she was 24 ± 6 years old, the grandmother gave birth to a daughter, who became the mother of LHA12 and LHA27. The grandmother also gave birth to a son who became the father of LHA34. Using the same logic, the son LHA12 became a father to LHA14 when he was 28 ± 7 years old. These generational intervals tightly constrain the probability distributions, which would otherwise be very large, mostly due to the uncertainty of the marine diet contribution and local ΔR. Taking all factors into account, model estimates the grandmother was born *~1280 (1240–1340, 95%)*.

Finally, one of the cousins (LHA27) is very similar to another sample (LHA23). They are both males who both died at ages 40–55 with strongly overlapping $^{14}$C ages, and $\delta^{13}$C and $\delta^{15}$N values. They were found in the same site and context, Las Huacas' Feature 17. This makes it possible that the samples are from the same individual; however, osteological analysis showed them to be two different individuals. Genomic data from LHA23 were contaminated, so they could not be used to evaluate this.

### Las Huacas: constraining dates based on stratigraphic relationships

At Las Huacas, stratigraphic relationships made it possible to build Bayesian models with depositional sequences. Here we refine a previous model for Room A2 in Complex N1, also called the Main Room[27]. We add details such as laboratory codes and additional context information (Supplementary Dataset 1c, d).

The precise starting boundary for the Main Room suggests when the site was first occupied, *~AD 1260 (1220–1280, 95%)*. Working from the bottom up, the main room's oldest and deepest dated contexts are from levels 10 and 11 (OS-149180, OS-149187, OS-149229). Next, there are four ages from levels 7, 8, and the hallway (OS-149185, OS-149182, OS-149226, and OS-149230). At this point, there are no more trenches with similar orientations. Above this, there is a set of kilns in level 6 (OS-149181, OS-149179, OS-149186). After the kilns were disassembled, the space was used by camelids (OS-149227, OS-149176). Next, the mortuary program in level 4 has terrestrial samples (OS-149188, UCIAMS-250737) and three individuals (LHA13/Ind 4, LHA190/Ind 8, and human hair from feature 34). The final phase has one terrestrial sample (OS-149183) and one individual (LHA67). The dates are older than expected based on their stratigraphic relationships (OS-149181, OS-149184, OS-149340), so we modeled them with a Charcoal Outlier model (Bronk Ramsey, 2009b), which resulted in acceptable agreement indices. One terrestrial sample (UCIAMS-183270) and one human tooth (LHA_Ind 3) could not be confidently placed in this stratigraphic sequence, so they are only included in the overall phase for these dates.

Feature 17 is one of the site's major deposits, a large ossuary with 11 dated individuals[26,28]. This context is mixed, and it is quite possible that bones were moved around, so we do not define a lower limit for this phase. All dated individuals were found below a single offering that was dated twice, on cotton and maize (OS-149177, OS-149178), with strongly bimodal calibration curves. Combining and modeling these dates shows the earlier peak is more likely, suggesting a precise upper limit for the burials below, *~AD 1490 (1460–1510, 95%)*.

### Middle Chincha Valley: burial dates with no stratigraphic relationships

In the middle Chincha Valley, research has produced 20 $^{14}$C ages from 10 burial contexts, with 1–3 contexts per site (for context details and

additional dates, see Bongers's dissertation[22]). These contexts were disturbed and mixed, so we cannot confidently build relationships into the Bayesian models. We group all the estimated death dates into a uniform phase that has imprecise boundaries of *~1260 (1180–1320, 95%)* and *~AD 1560 (1510–1650, 95%)*. The latest dates are from Tomb 1 (UC-008), which have calibrated medians of AD 1560 and 1580, but modeled as a phase and constrained by the models' ΔR, these medians shift to *~AD 1480* and *1520* (UCIAMS-155763, UCIAMS-155764).

The posterior probability distributions for the principal starting and ending boundaries are shown in Supplementary Fig 9.

### Agreement indices

Modeling all ages together, the agreement index is strong, $A_{model}$ = 150.2%, suggesting a robust model and coherence among the many factors built into the model. Nearly all dates have strong individual agreement indices. One posterior has <60% agreement index, for a maize sample (49.8%, OS-149178), but this is not a clear reason to manually exclude a date[101]. We retain it because we are confident in the priors that affect its probability distribution. Three diet-based calibration curve mixtures have low agreement indices, 42.5% for LHA13_Ind4, 68.7% for JUC73, and 63.5% for JUC35. This means that their dietary estimates may need to be reassessed. They could reflect short-term shifts in the ΔR, though there are also other possible reasons. JUC73 and JUC35 are the two earliest individuals in the sample, when ΔR may have been slightly different. Overall, there is strong agreement among the many individual mixed calibration curves, which suggests a coherent result for the ΔR. This is also reflected in its low error range, ±52.

Multiple runs resulted in minor differences, as expected given the stochastic nature of MCMC sampling[80].

### Summary kernel density estimates for individuals with similar ancestry and burial pigment

We used Kernel Density Estimates[102] to summarize the chronological trends of people with similar ancestries and burial patterns. We exported priors for death dates from the first model (see OxCal code), since this more closely reflects the timing of when the archeological context formed. We made three groups: deaths of people with only north coast ancestry (*n* = 11), people with mixed north and south coast ancestry (*n* = 11), and people who were treated with pigment after death (*n* = 16). All three groups include death dates from both the lower and middle valleys.

### aDNA laboratory work

All tooth samples from the 30 individuals investigated in this study were processed at the UC-PGL clean room facilities at the University of California, Santa Cruz, following strict precautions to prevent contamination[103]. DNA extracts were generated using a silica-column-based protocol optimized for the recovery of small aDNA molecules[104] with the addition of 0.2% bleach predigestion[105] and 50 mg of sample. All extracts were partially treated with Uracil-DNA Glycosylase (UDG) to reduce, but not eliminate, the amount of deamination-induced damage at the ends of the aDNA fragments[106]. Subsequently, we built double-indexed single-stranded DNA sequencing libraries from the extracts[107]. The libraries for all samples were first screened by sequencing 1 million reads for each on a NextSeq2000 (Illumina) for 2 × 150 cycles at UC-PGL. Libraries from samples showing sufficient DNA preservation were then enriched for 1,237,207 targeted SNPs across the human genome[29], using the TWIST Ancient DNA enrichment kit[30,31]. The enriched libraries were sequenced on several lanes of a NovaSeq X (Illumina) sequencer for 2 × 150 cycles at Fulgent Genetics (Temple City, CA). Negative controls were included with all extractions, library batches, and PCR amplifications.

## Sequencing read processing, chromosomal sex determination, screening and DNA authenticity

After demultiplexing, the resulting sequencing reads were processed using the in-house computational pipeline developed for aDNA described previously[108], available at (https://github.com/mjobin/batpipe). This pipeline merges paired-end reads (default parameters), maps sequencing reads against a user-specified reference genome, removes duplicate reads, and estimates quality traits. All shotgun-sequenced reads were mapped using BWA (v0.6.1)[109] against the human genome reference GRCh37/hg19. Mitochondrial DNA (mtDNA) reads were mapped against the human mtDNA reference rCRS[110].

Chromosomal sex was determined by evaluating the ratio (Ry)[111]. In addition, we employed a X-chromosomal normalization rate (Rx) approach that compares the Rx ratio to the variability observed in all autosomes[112].

We used the recommended parameters in Contammix[113] to estimate mitochondrial contamination rates and assessed contamination on the X-chromosome for all biologically male individuals using ANGSD[114]. We estimated patterns of DNA damage using Map-Damage 2[115].

## Mitochondrial and Y-chromosomal DNA analyses

The mitochondrial haplogroups of the individuals were determined using the HaploCart module implemented in the software vgan v3.1[116]. Since the used capture reagent did not include baits covering the entire mitochondrial (mt) genome sequence, the coverage for most samples was too low to reconstruct the complete mt-haplotypes. To determine the Y-chromosomal haplogroups of the male individuals, we used yHaplo[117], identifying the most derived allele upstream and the most ancestral allele downstream in the phylogenetic tree of the International Society of Genetic Genealogy version 15.73 (July 11th, 2020; http://www.isogg.org/tree).

## Population genetic analyses

We called genetic variants for all newly reported samples on the targeted 1240 k SNP positions, with a read chosen at random to represent this position using pileupcaller (https://github.com/stschiff/sequenceTools), after trimming 2 bp from each end of the reads using bamUTIL (https://genome.sph.umich.edu/wiki/BamUtil) to reduce potential bias introduced by DNA damage. The data were then merged with the previously published genomes from the middle Chincha Valley[15] as well as other previously published genome-wide data from ancient[33–36,118,119] and modern-day[39,120] South- and Central American individuals.

We used the qp3pop package in AdmixTools[41] to compute outgroup-f3-statistics with SEs calculated with a weighted block jackknife over 5-Mb blocks. Analyses were performed using the 1240 k SNP dataset. We used the inbreed: YES parameter to account for our random allele choice at each position. We generated a matrix of the outgroup-f3 values, converted these to distances by taking the inverse of the values and generated MDS plots.

We used the tools qpDstats, qpWave, and qpADM packaged in ADMIXTOOLS[41] to test for admixture. We computed several F4 statistics using the qpDstat (v970) package in ADMIXTOOLS using f4mode: YES, and printse: YES parameters. SEs were computed using a jackknife block size of 0.050. We used qpWave (v1200) from ADMIX-TOOLS to determine the minimum number of ancestry sources for each individual and groups of individuals using ancient and modern populations. We used the model parameters and outgroups (right populations) described in Bongers et al.[15] For all qpWave analyses, we used the default settings except for the change that we set allsnps: YES. For all individuals/groups where rank = 0 was rejected in the qpWave analyses, we used qpADM[41] to test two-way admixture models using the rotating model approach suggested as implemented in qpADM_wrapper (https://github.com/pontussk/qpAdm_wrapper, v1, 12 March 2022)[121]. We used a fixed set of sources/outgroups for the analysis, consisting of all ancient Andean ancestry clusters determined by ref. 33 and other ancient and modern-day populations from the Andes and the Amazon. We set the details: YES parameter, which reports a normally distributed Z score for the fit (estimated with a block jackknife).

## Relatedness and diversity

We used the software READv2[32] to determine first to 3rd degree inter-individual relatedness using default parameters, including all individuals that had ≥100,000 SNPs covered. To assess the genomic diversity of the studied groups, we performed conditional heterozygosity (CH) analyses using the Popstats package (https://github.com/pontussk/popstats; September 27, 2018), with the --pi flag and default settings otherwise[122]. To investigate runs of homozygosity we used the package hapROH[37], which allows us to infer ROH blocks from pseudohaploid ancient genomes. We included all samples with at least 400,000 SNPs covered in the 1240k target set in this analysis. The 1000 Genomes data were used as the reference dataset with the model parameter e_model = "haploid".

## Genetic data and dates from the cist graves

Two middle valley cist graves (Sector B Cist and UC-065A Tomb 3) located 4 kms apart (Fig S6), date to the 13th century and contain individuals with unadmixed north coast ancestry (JUC35 and JUC61). JUC35, from UC-065A Tomb 3, has a modeled median death date of ~AD 1290. JUC61, from the Sector B Cist, is not directly dated. The Sector B Cist contained at least five individuals (an adult male, an adult female, and three juveniles). A tooth sample (JUC73) from the same cist shares the same age and sex profile (adult male, 40–55 years) as JUC61. While not confirmed, it is therefore likely that JUC61 and JUC73 represent the same individual. JUC73 has a modeled median date of ~AD 1290. Two reeds from the Sector B Cist have modeled median dates of ~AD 1350.

## Reporting summary

Further information on research design is available in the Nature Portfolio Reporting Summary linked to this article.

# Data availability

All data needed to evaluate the conclusions in the paper are present in the paper and/or the Supplementary Information. Aligned sequencing reads for all individuals reported in this study are available from European Nucleotide Archive (ENA), accession no: PRJEB98110. Human remains and associated materials analyzed in this study are curated in Peru under the authority of the Peruvian Ministry of Culture. Specimens from the Chincha Valley (middle valley and Las Huacas) are housed in local archeological repositories, such as the Iea Regional Museum (museoregionaldeica@cultura.gob.pe), and remain under the care of the Peruvian Ministry of Culture.

# Code availability

Custom Mix-Cal-Lot script to translate estimated marine resource consumption output from MixSIAR to OxCal, and the full OxCal script used for this study, are available on Zenodo at https://doi.org/10.5281/zenodo.17917426 and in a .zip file attachment with this manuscript.

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

## Acknowledgements

The authors thank the communities of the Chincha Valley and the Peruvian Ministry of Culture for their support of this research. We appreciate the support from the Cotsen Institute of Archeology. Students and staff of the Chincha Archeological Field School made important contributions throughout the research process. Colleen O'Shea, Manuel Mamani, Natasha Vang and Tiffiny Tung assisted with sample selection and preparation. Paula Patricia Moreno Zapata and Rosa Estefita Zavaleta Barrios assisted in sample exportation. This research was funded by a National Geographic Early Career Grant and a University of California Multicampus Research Programs and Initiatives Catalyst Grant (UC-17-445724). A Rust Foundation Grant (RFF-2021-147) to JAD funded radiocarbon dates. Thanks to Dan Contreras, who wrote the R script that translates MixSIAR outputs into OxCal code. JLB acknowledges the NSF Graduate Research Fellowship under DGE-1144087, the Ford Foundation Fellowship Program, the National Geographic Young Explorers Grant Program under Grant 9347-13, and the Sigma Xi Grants in-Aid Research Program. JAD acknowledges financial support for excavations at Las Huacas provided by NSF DDRIG #1744365, Fulbright IIE, Lewis and Clark Award, and the University of Michigan. The American Museum of Natural History, Division of Anthropology provided support during the writing of this paper.

## Author contributions

Conceptualization: J.L.B.; J.A.D.; L.F.S. Methodology: J.L.B.; J.A.D.; L.F.S.; E.J.M.; J.R.R.; E.B.P.M.; J.O.; J.G.M.; K.K. Investigation: J.L.B.; J.A.D.;

A.R.Y.; I.A.S.; N.O.P.; L.F.S. Visualization: L.F.S.; E.J.M.; J.L.B.; J.R.R.; J.A.D. Writing—original draft: J.L.B.; J.A.D.; L.F.S.; E.J.M.; J.O.; J.R.R.; E.B.P.M. Writing—review & editing: J.L.B.; J.A.D.; L.F.S.; E.J.M.; J.O.; J.R.R.; E.B.P.M.; J.G.M;. K.K.; A.R.Y.; I.A.S.; N.O.P.

## Competing interests

The authors declare no competing interests.
