## [Transparent Peer Review file · Nature Communications]

Ancient DNA reveals a family ossuary and long-distance migration on the Pacific coast before the Inca Empire

Corresponding Author: Dr Jacob Bongers

Version 0:

Reviewer comments:

Reviewer #1

(Remarks to the Author)

This study combines radiocarbon dating, aDNA and dietary analysis to provide a new look into migration, kinship and intermarriage among groups living along the Pacific coast in Peru prior to the Inca empire. The data is combined in a Bayesian model using depositional sequences, genetic relationships and individually defined mixed marine and terrestrial curves. The individual curves were based on the percentage of marine carbon in the diet to provide more precise dates for the burials and a family ossuary. I do not have the expertise to evaluate the aDNA but will focus on the ^{14}C dating and modeling.

Because of the variability in upwelling off the west coast of Peru, radiocarbon values for humans consuming marine resources have not been able to be calibrated precisely. Rather than using a modern (pre-bomb) and highly uncertain value for ΔR , the offset of marine samples in the region from the global calibration curve, the authors derive ΔR within the Bayesian model.

The radiocarbon dating is sound but there is not enough detail on the OxCal model provided for the work to be reproduced. The complete OxCal model code needs to be provided in Supporting Information. This is essential to understand exactly what was done the chronological model, for instance I would like to know, among other things, how two unsampled (but genetically inferred) individuals were included in the chronological model (from Figure 2 caption Line 193-94). I would also like to see the probability plots for the OxCal queries and/or boundaries (e.g. 'End Las Huacas') rather than just the model intervals (as given in the spreadsheet 1b. Modeled Burial Dates). In addition, some documentation is needed for the Mix-Cal-Lot script used in conjunction with MixSiar dietary model. The OxCal code example given in supporting information code only shows the use of the histogram apparently resulting from the Mix-Cal-Lot script.

Specific comments:

Line 305-306: 'We estimate that the marine reservoir effect (ΔR) for this time and place was -314 ± 52 years.' It would be interesting to compare to the mean and standard deviation of the pre-bomb data.

In Supporting information: 'Marine ΔR for central Peru: constraining dates based on stratigraphic relationships'

'When using this database, basic queries only use one date from each location, since the database design is focused on spatial variation.' If there are more than one shell or coral series for a given location, they will be returned separately with a basic query, but for series it only returns the most recent collection year.

'In contrast, a full consideration requires manually calculating the ΔR of all data from each location.' This is not correct. The ΔR values for each sample in a series are given in the database but it does require accessing each series separately by clicking on the underlined map number in the table.

'The shells are all from the early twentieth century and only live a few years, but dates on a single shell can vary by centuries (Supplementary Figure 7)20–22.' Supplementary Figure 7 does not show dates on a single shell but rather seems to be the spread of all dates for the region.

Reviewer #2

(Remarks to the Author)

The submitted manuscript by represents a genomic study of individuals from two contexts in the Chincha Valley spanning two established cultural phases (the LIP and the Late Horizon) and with improved AMS dates that account for a marine reservoir effect.

In the interests of transparency, my areas of expertise pertain to the cultural history of the region and the archaeological contexts of the individuals studied; I am not a geneticist. With that caveat in place, I thought this manuscript grounded its interpretations of the individuals studied here within the larger corpus of genomic data for the Peruvian coast, and with the inclusion of more limited genomic data from the southern highlands. The authors' interpretation that most of the individuals were closely related and were part of consanguineal lineages strikes me as quite reasonable. Their interpretation that these groups were established on the North coast prior to the Incas' assimilation of the region also fits with archaeological and (limited) ethnohistoric research pointing to relative independence and cultural continuity. I also like that the authors interrogate the traditional LIP-LH boundaries; it is consistent with a larger shift in the archaeological work in the central Andes that is blurring this boundary considerably. Finally, I appreciate the attention to archaeological and ethnohistoric context in discussing the various motivations for people to move in (and presumably out) of the valley.

One question I have pertains to the generational timeline in Fig. 4. I am curious as to why the authors use 30 years to represent a generation (and how they estimated an average age at first birth of 24+/-6yrs for mothers and 28+/- for fathers). Is it because this would be the most conservative span, or is there another reason? Indigenous Andean age stages typically signify adulthood for both men and women well before then.

My other comments are minor and are embedded in the attached PDF.

Reviewer #3

(Remarks to the Author)

Dear authors, it's been a pleasure to read your manuscript about the Chincha Valley, genome wide data. It contributes to the comprehension of interactions and composition and diversity of the local populations. Kinship is a very important information for the archaeological understanding of precolumbian societies. It's noteworthy the effort of integrating genomic data to archaeological findings.

I understand that you have obtained all the permissions from the authorities and the consent of the local descendants, but none of the authors comes from Perú, although in the acknowledgements you mention several local students and staff that contributed in the manage and preparation of the samples, shouldn't be consider within the authors?

Data obtained is just before the INCA arrival and domination, their incorporation to this empire seems to have occurred without resistance from local inhabitants, which is in accordance with previous archaeological bibliography.

DNA was obtained and procedures and bioinformatics treatment were those recommended for ancient materials. Highly contaminated samples were discarded, also those exhibiting less than 20.000 SNPs or individuals with first- and second-degree relationships were excluded for the MDS.

Its originality is based, not only for the Chincha area but also because this is one of the first studies that analyzed red painted crania, and describes kinships among individuals studied in Las Huacas site, postulating the family reconstruction shown in figure 2. Mitochondrial lineages, were very important in this reconstructions of the LA family tree. It's interesting that B2b lineage appears earlier in the Pampas (<https://doi.org/10.1016/j.cell.2018.10.027>) and later in the very Patagonian south (<https://doi.org/10.1002/ajpa.24822>) Indicating earlier migrations southward. from the peruvian coast, this interesting information could be incorporated.

The methodology to analyze the result from genomic data is enough and adequate. When data will be available could be reproduced.

Results presented in the article support the conclusions. Although incorporation of new individuals from the middle Valleys may complete the information obtained.

I recommend publication including minor changes.

Version 1:

Reviewer comments:

Reviewer #1

(Remarks to the Author)

The authors have addressed my concerns adequately with the revision. It is really helpful to have the OxCal and Mix-Cal-Lot code.

I have only one additional comment which I think would provide a more fair critique of using shell data for Delta_R in archaeological contexts.

Supporting Material p6-7: 'In contrast, shell samples target annual or even seasonal growth. It seems that the temporal resolution mismatch means shell data are not appropriate for archaeological dates. This is salient in regions such as the Pacific coast with strong deepwater upwelling that can vary strongly by season and/or with strong El Niño activity.'

I understand the reasoning for the comment but I think it is too strong a statement about shell data not being appropriate for archaeological dates. I agree this is true in regions where there is strong and varying upwelling etc. but in other regions the pre-bomb shell dates show consistent Delta_R values with archaeological paired material several thousands of years older. Instead of saying the shell data are inappropriate I would suggest saying that shell data should be used with caution.

Otherwise I think the paper is ready for publication.

(Remarks on code availability)

I have reviewed the code but not in great detail. It code is well commented and easy to follow. Aa README file explains how to install and use the applications.

Reviewer #2

(Remarks to the Author)

Having read the authors' responses to my feedback and their revised manuscript, I recommend that the manuscript be accepted for publication.

(Remarks on code availability)

Reviewer #3

(Remarks to the Author)

Dear authors,

Thanks for considering my suggestion about the incorporation of Peruvian colleagues among the authors. Perhaps a deeper study of haplotype B2b in a comparative analysis would expand our knowledge about its origin and distribution in South America. But I understand it was not included in the aims of your paper. I believe the text has been significantly improved and is now ready for publication.

Best regards

(Remarks on code availability)

Responses to Feedback

We appreciate the insightful feedback from the reviewers. We have addressed their comments and improved the paper.

Reviewer #1:

The radiocarbon dating is sound but there is not enough detail on the OxCal model provided for the work to be reproduced. The complete OxCal model code needs to be provided in Supporting Information.

- *We have corrected this oversight and included the full OxCal code.*

This is essential to understand exactly what was done the chronological model, for instance I would like to know, among other things, how two unsampled (but genetically inferred) individuals were included in the chronological model (from Figure 2 caption Line 193-94).

- *We appreciate the reviewer asking about these individuals. This is based on the family tree. Since three sampled individuals have the same (unsampled) grandmother, we included her in the family tree. This is important because it allows us to incorporate spans between parents and children and hence spans between unsampled individuals. As described in the Supplementary Information we used data from a global 1948–2015 dataset on when males and females typically have children, and used these as priors in the model, as well as to create estimates for when the grandmother and daughter/mother would have lived. We also mention this in the OxCal code, which includes comments for clarity.*

I would also like to see the probability plots for the OxCal queries and/or boundaries (e.g. ‘End Las Huacas’) rather than just the model intervals (as given in the spreadsheet 1b. Modeled Burial Dates).

- *This is a good suggestion, and other readers may also prefer plots to a table. We have added these plots in Supplementary Figures 8 & 9.*

In addition, some documentation is needed for the Mix-Cal-Lot script used in conjunction with MixSiar dietary model.

- *Further description of what Mix-Cal-Lot does has been added to the Supplementary Information under section **Building mixed marine–terrestrial calibration curves with non-normal probability curves**. The entire custom Mix-Cal-Lot script and the raw MixSIAR input files are available in a Zenodo repository at DOI: 10.5281/zenodo.17917426 and in a .zip file submitted as an attachment. Both the Zenodo repository and the .zip file include a R Markdown version of Mix-Cal-Lot as well which provides further plain-language explanations of each step and visualizations of expected outputs.*

Line 305-306: ‘We estimate that the marine reservoir effect (ΔR) for this time and place was -314 ± 52 years.’ It would be interesting to compare to the mean and standard deviation of the pre-bomb data.

- *The 58 available dates from six shells are pre-bomb, collected between AD 1908 and 1948. Their weighted average ΔR is -15 ± 59 . It is very different from our Bayesian ΔR estimate. We clarified this in the Supporting Information.*

In Supporting information: ‘Marine ΔR for central Peru: constraining dates based on stratigraphic relationships’

‘When using this database, basic queries only use one date from each location, since the database design is focused on spatial variation.’ If there are more than one shell or coral series for a given location, they will be returned separately with a basic query, but for series it only returns the most recent collection year.

‘In contrast, a full consideration requires manually calculating the ΔR of all data from each location.’ This is not correct. The ΔR values for each sample in a series are given in the database but it does require accessing each series separately by clicking on the underlined map number in the table.

- *We thank the reviewer for catching this; we have clarified the text in the Supporting Information. We state the following: “When using this database, multiple dates from a single location can be accessed by clicking on the underlined map number in the table (note only one value per location is used by the site’s calculation option). For the 58 dates from this area, ΔR covers a wide range, from -181 to $+219$ years.”*

‘The shells are all from the early twentieth century and only live a few years, but dates on a single shell can vary by centuries (Supplementary Figure 7).’ Supplementary Figure 7 does not show dates on a single shell but rather seems to be the spread of all dates for the region.

- *Correct, this histogram shows the estimated ΔR for 58 dates on six shells from four locations. We clarified the figure caption.*

Reviewer #2:

One question I have pertains to the generational timeline in Fig. 4. I am curious as to why the authors use 30 years to represent a generation (and how they estimated an average age at first birth of 24 ± 6 yrs for mothers and 28 ± 6 for fathers). Is it because this would be the most conservative span, or is there another reason? Indigenous Andean age stages typically signify adulthood for both men and women well before then.

- *This is an excellent question. Cross-cultural estimates for generation length are between 25 and 30 years; we round to 30 in the timeline. In the OxCal models, we use*

precise numbers and error ranges. This is higher than ages that are often mentioned in informal discussions of generation length, which tend to focus on the age of parents at their first child, not all children. We have added a paragraph and citations clarifying this in the Supporting Information. The estimates we use are based on data derived from two global datasets that cover all countries for AD 1948–2015 (the Institute for Health Metrics and Evaluation and the United Nations Statistics Division). They agree with cross-cultural comparisons and genetic estimates. It is true that adulthood is well before this age, but here we are estimating the age of parents at all their children's births, not only the first child. We use this because, while aDNA can identify family relationships, we cannot know the order in which siblings were born.

Reviewer #3:

I understand that you have obtained all the permissions from the authorities and the consent of the local descendants, but none of the authors comes from Perú, although in the acknowledgements you mention several local students and staff that contributed in the manage and preparation of the samples, shouldn't be consider within the authors?

We thank the reviewer for raising this important point regarding the inclusion of Peruvian collaborators. Following a careful reassessment of contributions and in line with Nature Communications' authorship guidelines, we have updated the author list to include three Peruvian colleagues who played substantive roles in the research. Their contributions include coordinating research permits, overseeing local logistics, and performing bioarchaeological analyses essential to interpreting the results. All three have now reviewed and approved the manuscript and have agreed to be listed as co-authors.

We appreciate the reviewer's attention to equitable authorship practices and have ensured that our revised author list accurately reflects all individuals who made significant intellectual contributions to this work.

It's interesting that B2b lineage appears earlier in the Pampas(<https://doi.org/10.1016/j.cell.2018.10.027>) and later in the very Patagonian south (<https://doi.org/10.1002/ajpa.24822>) Indicating earlier migrations southward. from the peruvian coast, this interesting information could be incorporated.

We thank the reviewer for this observation. Sadly, the resolution of our mitochondrial data does not allow us to determine which subgroups of haplogroup B2b might be represented in our individuals. B2b is found in modern-day and ancient populations from Ecuador, the Central Andes, the Amazon, the Pampas, and Patagonia. We would need the complete mitochondrial genome resolution to determine the phylogenetic positioning of the lineages found in Chíncha and to assess their extent of relatedness, e.g., to those found in the Pampas. The general distribution pattern of B2b from Ecuador to the Southern Cone implies that the original lineage derives from the initial peopling, and not subsequent inter-population mobility.

Responses to Feedback

We appreciate the insightful feedback from the reviewers. We have addressed their comments and improved the paper.

Reviewer #1:

Supporting Material p6-7: 'In contrast, shell samples target annual or even seasonal growth. It seems that the temporal resolution mismatch means shell data are not appropriate for archaeological dates. This is salient in regions such as the Pacific coast with strong deepwater upwelling that can vary strongly by season and/or with strong El Niño activity.'

I understand the reasoning for the comment but I think it is too strong a statement about shell data not being appropriate for archaeological dates. I agree this is true in regions where there is strong and varying upwelling etc. but in other regions the pre-bomb shell dates show consistent ΔR values with archaeological paired material several thousands of years older. Instead of saying the shell data are inappropriate I would suggest saying that shell data should be used with caution.

- *We note that this approach can be effective in other regions, so it should be clear to readers that our comments are only meant for coastal Peru. We adjusted this sentence to say, "In contrast, the six modern shells from this region... are not appropriate for archaeological dates from coastal Peru" (p8).*
- *We adjusted this text following the reviewer's suggestions: "We recommend future models use modern shell data with caution, especially in Peru, and instead build floating Bayesian ΔR estimates based on archaeological priors." (p8)*
- *On page 7, we clarified that "... [using modern shells] can be effective in other regions."*